# Increasing Bioavailability of Trans-Ferulic Acid by Encapsulation in Functionalized Mesoporous Silica

**DOI:** 10.3390/pharmaceutics15020660

**Published:** 2023-02-16

**Authors:** Gabriela Petrișor, Ludmila Motelica, Denisa Ficai, Cornelia-Ioana Ilie, Roxana Doina Trușcǎ, Vasile-Adrian Surdu, Ovidiu-Cristian Oprea, Andreea-Luiza Mȋrț, Gabriel Vasilievici, Augustin Semenescu, Anton Ficai, Lia-Mara Dițu

**Affiliations:** 1Science and Engineering of Oxide Materials and Nanomaterials, Faculty of Chemical Engineering and Biotechnologies, University POLITEHNICA of Bucharest, Gh. Polizu 1-7, 011061 Bucharest, Romania; 2National Research Center for Food Safety, University POLITEHNICA of Bucharest, Splaiul Independentei 313, 060042 Bucharest, Romania; 3National Center for Micro and Nanomaterials, University POLITEHNICA of Bucharest, Splaiul Independentei 313, 060042 Bucharest, Romania; 4Department of Inorganic Chemistry, Physical Chemistry and Electrochemistry, Faculty of Chemical Engineering and Biotechnologies, University POLITEHNICA of Bucharest, Gh. Polizu 1-7, 011061 Bucharest, Romania; 5Academy of Romanian Scientists, Ilfov Street 3, 050044 Bucharest, Romania; 6National Institute for Research & Development in Chemistry and Petrochemistry—ICECHIM, 202 Spl. Independentei, 060021 Bucharest, Romania; 7Department Engineering and Management for Transports, University POLITEHNICA of Bucharest, 060042 Bucharest, Romania; 8Department of Microbiology and Immunology, Faculty of Biology, University of Bucharest, 1-3 Aleea Portocalelor, 060101 Bucharest, Romania; 9Research Institute of the University of Bucharest, 050095 Bucharest, Romania

**Keywords:** mesoporous silica, (3-aminopropyl)triethoxysilane, trans-ferulic acid, antimicrobial activity, controlled release

## Abstract

Two types of mesoporous materials, MCM-41 and MCM-48, were functionalized by the soft-template method using (3-aminopropyl)triethoxysilane (APTES) as a modifying agent. The obtained mesoporous silica materials were loaded with trans-ferulic acid (FA). In order to establish the morphology and structure of mesoporous materials, a series of specific techniques were used such as: X-ray Diffraction (XRD), Scanning Electron Microscopy (SEM), Brunauer-Emmet-Teller (BET), Fourier Transform Infrared Spectroscopy (FTIR) and thermogravimetric analysis (TGA). We monitored the in vitro release of the loaded FA at two different pH values, by using simulated gastric fluid (SGF) and simulated intestinal fluid (SIF). Additionally, *Staphylococcus aureus* ATCC 25923, *Escherichia coli* ATCC 25922, *Pseudomonas aeruginosa* ATCC 27853 and *Candida albicans* ATCC 10231 were used to evaluate the antimicrobial activity of FA loaded mesoporous silica materials. In conclusion such functionalized mesoporous materials can be employed as controlled release systems for polyphenols extracted from natural sources.

## 1. Introduction

As many bioactive substances can often exhibit unwanted effects at high concentrations, while only scanty drug concentrations with low efficacy are arriving at target tissue, various encapsulation methods and drug carriers have been developed [1,2].

Through the morphological control and the functionalization of inorganic nanoparticles, various advances in science have been developed so that delivery systems are more efficient in the transport of different drugs [3,4,5,6,7,8]. Mesoporous silica nanoparticles (MSNs) represent a new research opportunity for the development of controlled release systems [4,9]. The improvement of the transport systems of active substances in the body led to the development of new drug delivery systems. In the development of these systems, the goal is to keep the drug dose under control, a slow release, targeted release to the area of interest [10,11,12]. Due to the physico-chemical properties of mesoporous silica (very high pore numbers and specific surface), it allows high loading capacity, good compatibility and easy functionalization [13].

Starting from inorganic precursors, mesoporous silica can have different structures, such as MCM (Mobile Composition of Matter), SBA (Santa Barbara Amorphous), TUD (Technische Universiteit Delft), HMS (Hollow Mesoporous Silica) or MCF (Meso Cellular Form) [14,15]. The MCM family was the first type of mesoporous material synthesized since 1992, which include among other silica types, the hexagonal mesoporous MCM-41 and the cubic mesoporous arranged MCM-48 [16]. The major difference between the two materials is related to the arrangements of the pores. If MCM-41 has the pores arranges uniaxial, the MCM-48 has the pores arranged in a three-dimensional fashion and thus, the release of the active components will be unidirectional for MCM-41 or in all directions for MCM-48 [17].

Surface functionalization of mesoporous silica is one of the methods by which the interaction between drugs and mesoporous silica is improved for better efficiency in targeted delivery [18,19]. Covalent bonding between functional groups and silica can be achieved by grafting and co-condensation, post-synthesis and direct incorporation methods [20,21,22]. The most common functionalization methodology of the mesoporous silica surface are made with groups such as: amine (3-aminopropyl)triethoxysilane, thiol (3-mercaptopropyl)trimethoxysilane and sulfonic acid (trimethyl[propyl]ammonium chloride) [23,24,25,26].

The structure of mesoporous silica allows it to be loaded with drugs, polyphenols or more complex natural agents such as natural extracts [27,28,29]. Polyphenols are given special attention due to their pharmacological properties and because they are extremely numerous found in various plants, fruits and vegetables [30,31,32,33,34] while their biological activity is very wide. The porous structure of the silica-based materials allows a high loading capacity with various drugs or bioactive substances (e.g., antimicrobials, polyphenols) and further facilitate the controlled release by gradual unloading into the target tissue. By achieving this slower release of the drugs from the pores of the silica particles, a steady concentration of the released substance can be maintained, ensuring a reliable biological activity for longer times [35]. Due to the proven potential of biologically active agents, such as specific polyphenols, in preventing or mitigating the onset of chronic diseases, considerable interest has been shown in the development of dietary supplements containing polyphenols and/or consumer products rich in polyphenols [36,37].

Polyphenols are extremely important components of the human diet due to their special properties such as antioxidant activity, free radical scavenging abilities and the ability to function as antibiotics (exerting anti-diarrheal, anti-ulcer, and anti-inflammatory effects) and to alleviate tissue damage induced by oxidative stress associated with chronic diseases [38,39,40].

Ferulic acid (FA) is part of the phenolic acid class and is a derivative of cinnamic acid frequently found in numerous plants [41,42]. One of the most important roles of FA is that it has a strong antioxidant activity, but it has multiple pharmacological activities such as antimicrobial, antibacterial, anticancer, anti-inflammatory, antidiabetic, etc. [43,44,45,46,47,48]. Ferulic acid is a powerful antioxidant ingredient found in many vegetables, fruits and grains. It absorbs UV rays and protects the skin from the harmful effects of ultraviolet radiation, especially erythema and skin irritations. If it is used together with vitamin C and vitamin E, it potentiates the antioxidant effects, the removal of their pigment spots and the stability of the cosmetic product formula.

The aim of this study was to synthesize mesoporous materials, namely MCM-41 and MCM-48, and to functionalize them by grafting with amino groups following the reaction with (3-aminopropyl)triethoxysilane (APTES). The obtained materials were loaded with trans-ferulic acid to observe the loading capacity of mesoporous silica. After obtaining all the materials, they were investigated to determine the pore volume and surface area with Brunauer-Emmet-Teller analysis and the structure of the materials with X-ray diffraction (XRD). The morphology of the synthesized, functionalized and loaded materials was analysed through Scanning Electron Microscopy (SEM), and the identification and estimation of the loading quantity with trans-ferulic acid was determined by Fourier Transform Infrared Spectroscopy (FTIR) in connection with thermogravimetric analysis (TGA). The materials loaded with trans-ferulic acid were subjected to a in vitro release study in two simulated biological fluids with different pH to observe their release profiles. The FA loaded mesoporous silica materials were tested to evaluate the antimicrobial activity on four strains *Staphylococcus aureus* ATCC 25923, *Escherichia coli* ATCC 25922, *Pseudomonas aeruginosa* ATCC 27853 and *Candida albicans* ATCC 1023. The natural agents, such as polyphenols, could be good candidates in the treatment of infections especially when the infections involve resistant bacteria and common antibiotics are not efficient. Moreover, these silica particles loaded with polyphenols have an additional advantage because the residues (faces) does not generates microbial resistance.

## 2. Materials and Methods

### 2.1. Materials

Mesoporous silica was prepared starting from tetraethyl orthosilicate (TEOS) as silica precursor, cetyltrimethylammonium bromide (CTAB) as template agent, ammonia (NH_3_) and absolute ethanol, all from Merck (Darmstadt, Germany). According to the ratio between the components, two mesoporous systems (MCM41 and MCM48) were obtained as we previously presented in [35,49]. In this paper, a special attention was paid to the influence of the functionalisation and thus, both types of MCM materials were functionalized with (3-aminopropyl)triethoxysilane (APTES) purchased from Sigma Aldrich (St. Louis, MO, USA), while the loading of the mesoporous materials was done with trans-ferulic acid (FA, ≥98%) solution in acetone (both from Sigma Aldrich).

Release study was performed in simulated intestinal and gastric solutions obtained by mixing sodium chloride (NaCl) purchased from Sigma Aldrich, sodium hydroxide (NaOH, 1 N), hydrochloric acid (HCl, 2 N) purchased from S.C. Silal Trading SRL (București, Romania) and potassium dihydrogen phosphate (KH_2_PO_4_, ≥98%) from Carl Roth (Karlsruhe, Germany). 

All the substances were used without further purification in all performed experiments together with distilled water (dw).

All strains tested in this study were obtained from the Microorganisms Collection of the Department of Microbiology, Faculty of Biology & Research Institute of the University of Bucharest.

### 2.2. Equipment

Mesoporous materials, of MCM-41 and MCM-48 types, and APTES functionalized mesoporous silica, as well as the trans-ferulic acid loaded mesoporous samples, were characterized by modern, usual techniques. X-ray powder diffraction (XRD) diffractograms were recorded by using Cu_Kα_ radiation with a Panalytical X’Pert Pro MPD instrument. A Thermo IN50 MX instrument was used to record the FTIR spectra in attenuated total reflectance mode (ATR) for all the samples. The recorded spectra were used to carried out the study of the structural features of the MCM supports as well as the drug delivery systems. BET analysis (the N_2_ adsorption isotherms) was performed on a NOVA 2200e Gas Sorption Analyzer (Quantachrome, Boynton Beach, FL, USA), at 77.35 K and relative pressure *p*/*p*_0_ = 0.005–1.0. Before measurements, the materials were outgassed for 4 h at 110 °C under vacuum. The surface morphology of the samples was examined using a QUANTA INSPECT F electron microscope equipped with a field emission gun and an energy dispersive (EDS) detector, on samples covered with silver. Thermogravimetric analyses were recorded using a Netzsch 449C STA Jupiter instrument in the 20–900 °C temperature interval, in an open crucible, under a dynamic atmosphere of dry air (50 mL/min). A heating rate of 10 °C/min was employed.

The controlled release of trans-ferulic acid from the obtained mesoporous materials was evaluated with an Agilent 1260 Infinity High Performance Liquid Chromatograph equipped with Diode Array Detector (HPLC-DAD). Ultrapure water and acetonitrile (HPLC grade, Sigma Aldrich) were used as mobile phase and separation was performed on an Aqua C18 column (250 × 4.6 mm, 5 μm).

### 2.3. Preparation and Functionalization of Mesoporous Materials

Both mesoporous materials, MCM-41 and MCM-48, used as a support for functionalization and further loading, were synthesized using the soft template method starting from proper ratio of TEOS and CTAB (Figure 1). The synthesis was carried out under the same conditions, the working conditions being described in our previous work [49].

The functionalization of mesoporous silica was done following the method [50], using (3-aminopropyl)triethoxysilane as coupling agent. One gram of each mesoporous material (MCM-41 and MCM-48), was dried in the vacuum drying oven at a temperature of 23 °C for 3 h. After drying, 100 mL ethanol was added and the suspensions were sonicated for one hour. 125 μL of APTES was added to each suspension and left to reflux for 24 h and sonicated for one hour. The obtained suspensions were filtered and left to dry.

### 2.4. Adsorption of Trans-Ferulic Acid

To load the obtained mesoporous materials, a saturated solution of 0.4 g of trans-ferulic acid (FA) in 4 mL of acetone was prepared. 1 g of mesoporous material was added in a glass vessel and kept under vacuum, 2 mbar, for 30 min. After this period of time, the saturated solution of trans-ferulic acid was added stepwise (in three successive steps) to enter the pores of the material. After drying at 60 °C mesoporous materials loaded trans-ferulic acid were obtained as presented in Table 1. In all cases, the FA was loaded in a 0.4:1 ratio (*w*/*w*) to the mesoporous support materials.

### 2.5. In Vitro Release Study

The release profiles of the materials loaded with trans-ferulic acid were evaluated in two simulated biological fluids, simulated gastric fluid (SGF, pH = 1.2) and simulated intestinal fluid (SIF, pH = 6.8) [35,51]. The release of the trans-ferulic acid from the mesoporous materials (functionalized or not with APTES) was done in SGF and SIF (140 mL) solution considering 50 mg mesoporous supports loaded with trans-ferulic acid. Throughout the study, the solutions were kept under magnetic stirring at 37 ± 2 °C. Samples were taken at predetermined time intervals and FA concentration was quantified by high performance liquid chromatography.

### 2.6. Antimicrobial Activity

*Staphylococcus aureus* ATCC 25923, *Escherichia coli* ATCC 25922, *Pseudomonas aeruginosa* ATCC 27853 and *Candida albicans* ATCC 10231 were used to evaluate the antimicrobial activity of silica mesoporous materials.

#### 2.6.1. Quantitative Determination of Antimicrobial Activity

The minimum inhibitory concentration (MIC) evaluation was made by using the decimal microdilution method in a liquid medium: Nutrient Broth for bacterial species and Sabouraud for fungal species. In this sense, suspensions of microbial cells are made in a sterile physiological buffer (PBS) using 18–24 h cultures, reaching a standard density of 1.5 × 10^8^ CFU/mL (colony forming units/mL) for bacterial species, respectively 3 × 10^8^ CFU/mL for *Candida* sp. Decimal dilutions are made from each nanoparticle suspension, followed by inoculation with a standard microbial suspension (medium liquid volumetric ratio: microbial suspension = 10:1). Following the same steps, the blank/control (C) samples are performed: sterility and microbial growth. The 96-well plates are incubated at 37 °C for 24 h. The determination of MIC values was performed both by macroscopic observation and by reading at 620 nm (the suspensions are transferred to a sterile plate because the initial plate contains precipitate/NPs deposited on the bottom of the well and would interfere with the absorbance of the medium) [52,53,54].

#### 2.6.2. Semi-Quantitative Evaluation of Microbial Adherence to an Inert Substratum

The evaluation of the inert substrate adhesion potential of the tested strains was carried out by the crystal violet staining method. After 24 h of incubation, 96-well plates are washed thrice with phosphate buffered saline solution and fixed with cold CH_3_OH (5 min). After its removal, the dried plates are stained with 1% crystal violet solution (15 min). The excess dye is removed by washing, and in order to determine the MAIC values (minimum adhesion inhibition concentration), the dye included in the cells adhered to the walls of the well is dissolved with 33% acetic acid. Spectrophotometer readings will be performed at 490 nm with the BioTek Synergy™ HTX ELISA Multi-Mode Reader (BioTek, Winooski, VT, USA) [52,53,54].

## 3. Results and Discussions

All the materials, including pure supports as well as the loaded mesoporous materials were characterized by the appropriate physico-chemical methods.

### 3.1. X-ray Diffraction

All four diffraction patterns recorded at low angles on the mesoporous MCM-41 materials, Figure 2, present the typical arrangement for these materials. The strong diffraction peak from 2.43°, is attributed to the (100) crystallization plane, while the other less intense peaks correspond to the (110), (200) and (210) crystallization planes, characteristic to the hexagonal mesoporous MCM-41 structure. These peaks are consistent with literature data [55].

From the XRD data it can be seen that the functionalization of MCM-41 with APTES does not alter the ordered mesoporous structure of silica, but a slight shift and decrease in the intensity of the peak feature of the (100) crystallization plane is observed [56,57]. This is due to the formation of a silica layer bearing aminopropyl moieties on the surface and in the pores of the mesoporous material as a result of the hydrolysis and condensation reactions that take place during the functionalization reaction. At the same time, the pores of the mesoporous material decrease in size and this is also visible in XRD.

In addition, it is observed that the loading of MCM-41 and MCM-41_APTES materials with trans-ferulic acid, carried out under vacuum, does not alter the ordered mesoporous structure of the silica, but induces a considerable decrease in the intensity of the peaks and alters the shape of the peak characteristic for the (100) crystallization plane, due to the absorption of trans-ferulic acid. Due to the specific pattern of the XRD spectra it is expected that FA is mostly loaded inside the pores [35].

The X-ray diffractograms obtained for MCM-48 based materials, Figure 3, present the characteristic (211) peak at 2.52°, while the other three important peaks, (220), (420) and (332) are positioned at higher 2Θ values. This arrangement is characteristic to the cubic structure of MCM-48 [58]. The X-ray diffraction data obtained for MCM-48 materials functionalized with APTES, indicates a slight shift of the peak corresponding to the (211) crystallization plane [57,59] due to the formation of a silica layer on the surface and in the pores of the mesoporous material as a result of the reactions of hydrolysis and condensation occurring during functionalization reaction. At the same time, the pores of the mesoporous material decrease in size.

For the mesoporous materials that were loaded with ferulic acid, the XRD data indicates a considerable decrease in the intensity and the degree of crystallinity, in addition to the shift of the (220) and (211) characteristic peaks. All these information, the decrease in intensity, the crystallinity degree and the characteristic peaks shifting, may indicate the loading of polyphenols in the pores of the mesoporous material, but also a strong interaction between the two components.

### 3.2. Specific Surface Area—Brunauer-Emmet-Teller Adsorption Isotherms

Following the functionalization process of the mesoporous material MCM-41/48 with APTES, it can be seen from BET data (Table 2) that the specific surface area slightly decreases from 1365 m^2^/g for MCM-41, to 1014 m^2^/g for the material MCM-41_APTES and from 1582 m^2^/g for MCM-48, to 1555 m^2^/g for the material MCM-48_APTES as a result of the deposition of APTES on the surface of the mesoporous nanoparticles, which causes the material’s pores to become narrower as a consequence of the functionalization. The overall surface area of these mesoporous materials has two components, the external surface area of the spherical particles, with a small contribution—less than 1 m^2^/g, and the internal surface area of the cylindrical pores, which assure the very high surface area. Because during the loading process, the FA will enter inside the pores of the material, filling them, the BET determined specific surface area will drastically decrease, roughly by a factor of 3. As a consequence of the FA loading, the pore size decrease, and some of them even disappear when the loading degree is high, therefore, the overall specific surface area decreases considerable.

In the case of MCM-41_APTES, the functionalization with APTES leads to a decrease of the determined BET surface area of 1.34 times, while after FA loading, in the case of MCM-41_APTES_FA the decrease is 3.36 times. In the case of MCM-48, a higher BET surface area is observed compared to MCM-41, but after functionalization the BET surface area for MCM-48_APTES slightly decreases (by 1.01 times). After loading with FA the BET surface area for MCM_48_APTES_FA decreases by 3.08 times.

Analysing the data from Table 2, we can conclude that the BET data are in good agreement with the XRD analysis, indicating that the FA is adsorbed inside the pores of the mesoporous materials.

### 3.3. Fourier Transform Infrared Spectroscopy

In order to characterize the structure, the FT-IR spectra of MCM-41, MCM-41_FA, MCM-41_APTES and MCM-41_APTES_FA (Figure 4). In the FTIR spectra of MCM-41 mesoporous materials the most important peaks of characteristic vibrations can be identified. The broad bands between ~3200–3400 cm^−1^ correspond to the associated silanol (Si-OH) surface groups and adsorbed hydrogen-bonded water molecules. The bands at 1239 cm^−1^ and 1057 cm^−1^ can be assigned to the stretching vibrations of the asymmetric Si-O-Si units, while the band from ~980 cm^−1^ is associated with silanol groups of MCM-41. The stretching vibrations of the symmetric Si-O-Si units can be identified as the band at 807 cm^−1^ and the peak from 440 cm^−1^ is generated by the Si-O-Si moiety’s deformation vibrations [60,61].

The characteristic peaks for Si-O-Si bonds from the APTES functionalized mesoporous material, shifted to slightly higher wavelengths, can be identified in the FTIR spectrum (Figure 4). In addition, the bands characteristic of the functional groups in the APTES structure are present. The presence in the FTIR spectra of the band at 2980 cm^−1^, characteristic of the three methylene (propyl) groups clearly proves the functionalization of MCM-41.

Unfortunately, the bands associated with NH_2_ cannot be identified because of the lower molar absorptivity and lower content. Additionally, the bending vibration bands characteristic of the N–H bond normally present around 714 cm^−1^, the symmetric bending vibration band characteristic of the –NH_2_ group and the stretching vibration band corresponding to the C–N bond normally present around the value of 1000–1200 cm^−1^ cannot be highlighted as a result of the overlap with the vibration/stretching bands characteristic of Si–O–Si bonds in the range 1000–1130 cm^−1^ and that of Si–CH_2_ stretching in the range of 1200–1250 cm^−1^ [58].

By loading MCM-41 and MCM-41_APTES with trans-ferulic acid, a series of new bands appear in the FTIR spectra (Figure 4), between 1300–1800 cm^−1^, being associated with the presence of FA into the mesoporous material. The peaks from 1684 cm^−1^ and 1700 cm^−1^ are associated with the stretching vibrations of the functional groups C=O and –OH, from FA. The peaks from ~1360 cm^−1^ and 1629 cm^−1^ (for MCM-41_FA) and 1387 cm^−1^ and 1630 cm^−1^ (for MCM-41_APTES_FA) are assigned to the stretching vibration of the bond between the aromatic nucleus and the carboxyl moiety in the FA [35]. The observed shifts can be explained considering the interactions which appears between the support and trans-ferulic acid.

In the FTIR spectra recorded for the MCM-48 type mesoporous materials (Figure 5), one can also observe the most important vibration peaks, characteristic for the Si-O-Si network, at ~1239 cm^−1^, 1054 cm^−1^, 978 cm^−1^, 810 cm^−1^ and 436 cm^−1^, the other features being similar with those of MCM-41 corresponding materials.

### 3.4. Scanning Electron Microscopy (SEM)

SEM micrographs (Figure 6) recorded for the mesoporous materials based on MCM-41 indicate that particles have variable size of hundreds of nanometres and a spherical morphology. When compared to the bare MCM-41, SEM images of MCM-41_FA and MCM-41_APTES_FA show some minor heterogeneities on the external surfaces of the mesoporous silica spheres, which can be attributed to the deposition of ferulic acid. Considering their incidence, and low content, it can consider that the ferulic acid (FA) is mostly deposited into the pores of the mesoporous materials. At high magnification (100,000×), it could be clearly observed that the SEM images of the MCM-41 sample were more translucent compared to the samples modified with APTES or loaded with ferulic acid (FA). These observations are in good agreement with the XRD and FTIR data, but also with the BET analysis [35].

SEM micrographs (Figure 7) recorded for the mesoporous materials based on MCM-48 indicate the presence of quasi-spherical silica particles with sizes in the 150–400 nm interval. In addition, similarly to MCM-41 based samples, we can consider that FA is mainly loaded inside the pores.

### 3.5. Thermogravimetric Analysis

The thermal stability of the samples, both consisting of mesoporous materials and materials loaded with polyphenols, was studied by thermogravimetric analysis. Firstly the TG curves for MCM-41 (Figure 8) and MCM-48 (Figure 9) permits the determination of some parameters like surface density of adsorbed H_2_O molecules or -OH moieties, calculated as indicated in [35,49] (Table 3). The recorded mass loss up to 200 °C (1st mass loss) is generated by the elimination of physically adsorbed H_2_O molecules, from particles surface or from inside the pores. The 2nd mass loss, in the interval 200–900 °C is generated by the condensation of silanol moieties (Si-OH) which leads to the silica densification.

The successful functionalization of the MCM samples with APTES is indicated by the different TG/DSC curves shape for the functionalized mesoporous materials MCM-41_APTES (Figure 8) and MCM-48_APTES (Figure 9) vs. bare support. The sample are presenting a small mass loss up to 305 °C because of elimination of water molecules adsorbed on the surface of nanoparticles and condensation of –OH moieties. The process is similar with that of bare MCM-41 or MCM-48, but with higher intensity, indicating a better adsorption capacity. APTES has a boiling point of 217 °C but no thermal effect is recorded up to 300 °C, indicating the successful bonding between MCM and APTES. After 305 °C the samples are suffering a degradation process, with the main mass loss being recorded up to ~700 °C. This is an oxidation process of the organic part as indicated by the strong exothermic effects with peaks at 313 and 309 °C respectively.

The MCM-41_FA and MCM-41_APTES_FA samples present a weak endothermic peak with onset at 170.3 and 168.2 °C respectively, assigned to the melting of loaded FA. Same type of endothermic effect is present on DSC curve of MCM-48_FA sample, with onset at 170.3. For the MCM-48_APTES_FA sample the effect is not visible, only a small inflexion point at 144.8 °C being detected. The lack of a clear melting effect and the smaller temperature recorded as an inflexion on DSC curve for the MCM-48_APTES_FA sample indicates stronger interactions between FA and the mesoporous support in this case. The relevant data are presented in Table 4.

The results indicate that the APTES functionalization has an impact especially on estimated load on MCM-48_APTES_FA vs. MCM-48_FA, decreasing the FA loaded amount from ~32 to ~26%.

### 3.6. In Vitro Release Study

Release curves of FA from mesoporous materials and functionalized mesoporous materials are shown in Figure 10. As can be seen, all unmodified mesoporous materials loaded with FA had a fast release profile of up to 70–80% (within six hours) compared to mesoporous materials functionalized with amino groups and loaded with FA that had a slower release profile of up to 40–60% (within 24 h) [35]. From this difference between the release profiles, it can be clearly observed that the functionalization with aminopropyl groups considerably reduced the release rate of the active substance. Considering also, our previous works [35,49] it is obvious that the chemical surface modification with APTES or other silanization agents can be a good solution to induce a better tuning capacity of the release of these polyphenols for specific applications, including food supplements. As can be seen in the study, the release rate depends on the interactions of the functional group with the substance, which means that the use of different silanization agents bearing (-NH_2_, -COOH, -SH) could be exploited in tuning the delivery characteristics of the trans-ferulic acid [62]. The functionalization of the mesoporous materials with aminopropyl moieties has an important impact on both the release profile and rate over the first six hours. It can see that the release is slightly increasing over this period of time and this can be, most probably explained by the development of strong hydrogen bonds between the NH_2_ (amino moieties of the MCM support) and OH and COOH (of trans-ferulic acid)—also visualized in FTIR as a ~28 cm^−1^ shift of a specific band of COOH. Depending on the pH, even electrostatic interactions can appear, amino groups being susceptible to protonation, especially at pH < 7.2 while carboxyl groups are susceptible to deprotonation at pH > 7. As indicated by the experimental data presented in [63] the normal gastrointestinal transit time can be divided in three intervals: first 2–5 h are spent in the stomach, followed by 2–6 h of release in small intestine and 10–60 h in the large bowel. By coupling these times with the data obtained from release profiles in SGF and SIF, we can state that between 20–40% of FA will be released in the stomach (depending on actual transit time), with additional 20–40% of loaded FA being release in the small intestine. Additional FA quantities can be released in large bowel as a function of the actual transit time. In addition, after FA loading, the APTES functionalized MCM-41 or MCM-48 particles can be embedded in a mucoadhesive system [64,65], that will allow modulation of residence time and FA recovery [66]. Furthemore, by comparing with antimicrobial drugs, such as antibiotics, the negative environmental impact is low as these systems don’t lead to antimicrobial resistance.

In both, SGF and SIF, even after 24 h, the cumulative release of FA from the silanized mesoporous samples remain still below that from the bare mesoporous systems at 6 h, proving that aminopropyl groups have an important impact over the release rate. In SGF, the delayed nature of the delivery of FA is also observed but the difference between the cumulative releases of FA from the silanized samples comparing with the bare mesoporous samples is lower.

By comparing the MCM-48_FA with MCM-48_APTES_FA system, the first observation is that the release curves are similar in both simulated biological fluids, SGF and SIF. Unlike the systems containing MCM-48, the systems based on MCM-41 present a different behaviour. MCM-41_FA, had a faster release profile in the simulated intestinal fluid while and MCM-41_APTES_FA had a slower release when compared to MCM-48 materials.

### 3.7. Antimicrobial Activity

The mesoporous silica materials were tested to evaluate the antimicrobial activity on four strains *Staphylococcus aureus* ATCC 25923, *Escherichia coli* ATCC 25922, *Pseudomonas aeruginosa* ATCC 27853 and *Candida albicans* ATCC 1023.

#### 3.7.1. Quantitative Evaluation of Antimicrobial Activity

The lowest concentration at which the tested samples still can inhibit the microbial growth represents the minimum inhibitory concentration (MIC). The obtained values (mg/mL) are presented in Table 5.

The quantitative results for the silica-based mesoporous materials indicated the highest sensitivity against *P. aeruginosa* which is an important observation, considering that this species is frequently involved in healthcare associate infections. It can be observed that the materials loaded with ferulic acid negatively influence the growth of pathogens strains tested, compared to the controls. The MCM-41_APTES_FA and MCM-41_APTES samples determined the highest sensitivity of *S. aureus*, *P. aeruginosa* and *C. albicans*, with MIC values ranging from 0.01 and 0.001 mg/mL. The MCM-41_APTES_FA presented the best antimicrobial activity against Gram-positive and Gram-negative bacteria tested in this study. Overall, a synergic effect between mesoporous materials and ferulic acid can be observed.

A recent study [67] also observed the bacteriostatic effect of ferulic acid against Gram-positive and Gram-negative bacteria. Also, Borges et al. [45] determined the antibacterial activity of ferulic acid against *S. aureus*, *E. coli*, *P. aeruginosa* and *L. monocytes*. The MIC values are 1 mg/mL for *E. coli* and *P. aeruginosa*, and 1.25 mg/mL for *S. aureus*. Ferulic acid produced changes in membrane properties, especially hydrophobicity changes, decreasing negative charge and damage to molecular mechanisms.

#### 3.7.2. Semi-Quantitative Assessment of Microbial Adherence to the Inert Substratum

In Table 6 are presented the minimum inhibition concentrations (mg/mL) for adherence to the inert substratum, for the silica-based mesoporous materials.

As indicated by the MIC and MAIC values (Table 5 and Table 6), *S. aureus*, *E. coli* and *P. aeruginosa* are the most sensitive strains. The lowest values of MAIC range from 0.001 and 0.1 mg/mL in the case of silica-based mesoporous materials. Among all the studied samples, MCM-41_APTES_FA prevented the adherence of pathogen strains to the inert substratum, respectively, the development of the mature and stable biofilm, and presenting the most pronounced inhibitory effect. The samples functionalized with APTES and/or loaded with ferulic acid present a better bacteriostatic effect on the strain’s growth than mesoporous silica. Accordingly, with the previous assays, it seems that ferulic acid not only determined the damage to cell growth but also MCM-41/MCM-48 functionalized with APTES induced the hydrophobicity changes and the bacteriostatic effects. Furthermore, ferulic acid, as a dietary polyphenol, improves the modulation of gut microbiota [68,69] and the antimicrobial activity of silica-based mesoporous materials loaded with ferulic acid and, in vitro release profiles, confirms the potential for these materials as food supplements. In future, we want to continue this study with biocompatibility assay and the influence of these materials on probiotic bacteria/intestinal microbiota.

## 4. Conclusions

In this study, two types of functionalized mesoporous materials, MCM-41 and MCM-48, were synthesized by the soft-template method using (3-aminopropyl)triethoxysilane (APTES) as a modifying agent. The BET, TGA and XRD analyses indicate that mesoporous materials were obtained and further functionalized with APTES. The as obtained functionalized mesoporous materials were loaded with trans-ferulic acid using the vacuum assisted loading technology and found that the loading mainly occurs within the pores. The APTES functionalized mesoporous materials exhibited comparatively loading capacity when compared with the simple MCM-41, but smaller values for MCM-48. Strong interaction between the functionalized support and trans-ferulic acid are proved by a strong FTIR shift of about 28 cm^−1^ of the carboxyl group, and lower melting temperature of the loaded FA. Due to these strong interactions, the release rate of FA decreased significantly in both SIF and SGF, which can be beneficial allowing the development of sustained drug delivery systems of biological active agents, including polyphenols, for antimicrobial, antibacterial, anticancer, anti-inflammatory or antidiabetic purposes. The use of ferulic acid, instead of a specific antibiotic, has the advantage that the release into the nature of the residual polyphenol (FA not released within the gastrointestinal tract) will not have a negative environmental impact.

## Figures and Tables

**Figure 1 pharmaceutics-15-00660-f001:**
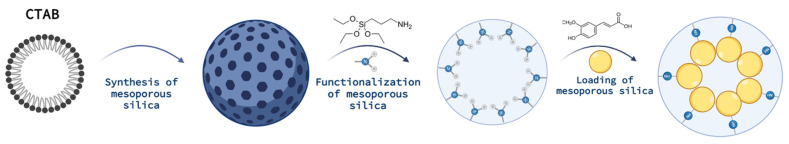
Synthesis of functionalized mesoporous systems loaded with ferulic acid (Created with BioRender.com).

**Figure 2 pharmaceutics-15-00660-f002:**
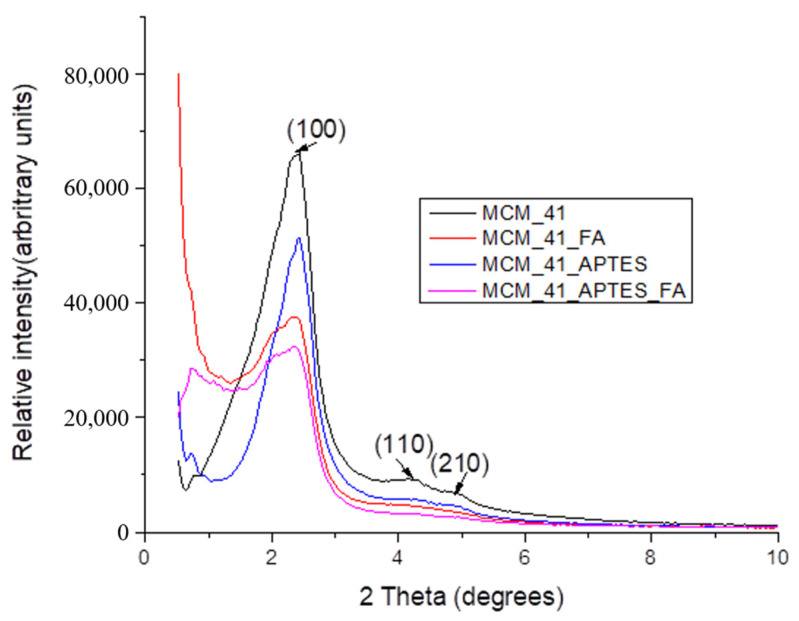
X-ray diffractograms for the MCM-41 based samples: MCM-41, MCM-41_FA, MCM-41_APTES and MCM-41_APTES_FA.

**Figure 3 pharmaceutics-15-00660-f003:**
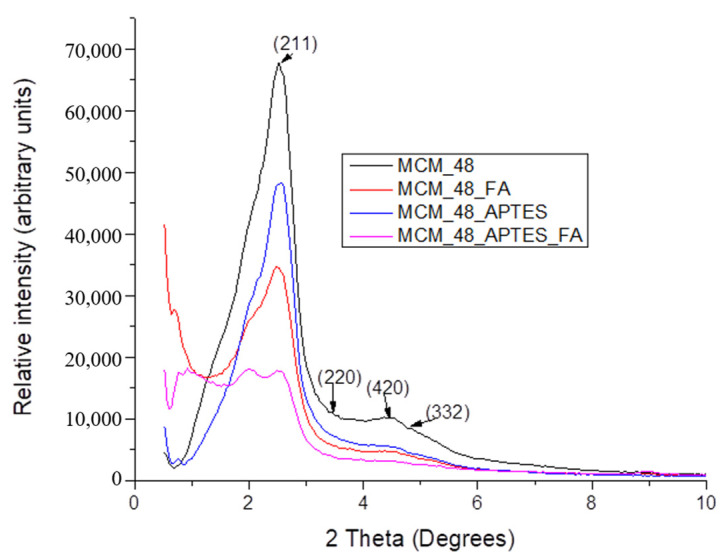
X-ray data for the MCM-48 based samples: MCM-48, MCM-48_FA, MCM-48_APTES and MCM-48_ APTES _FA.

**Figure 4 pharmaceutics-15-00660-f004:**
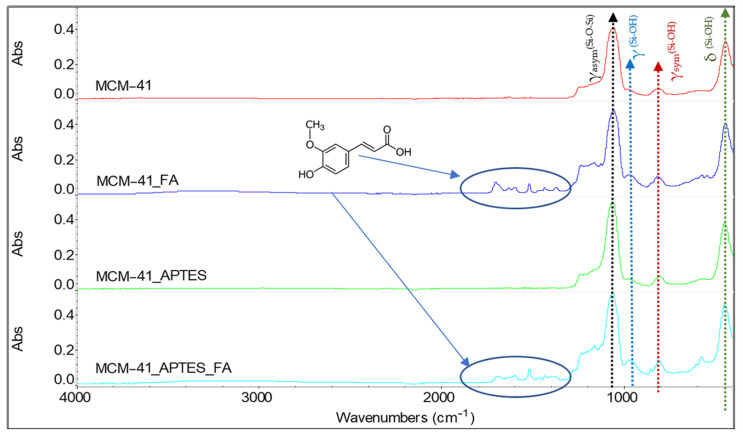
FTIR spectra and band assignments for the samples MCM-41, MCM-41_FA, MCM-41_APTES and MCM-41_APTES_FA.

**Figure 5 pharmaceutics-15-00660-f005:**
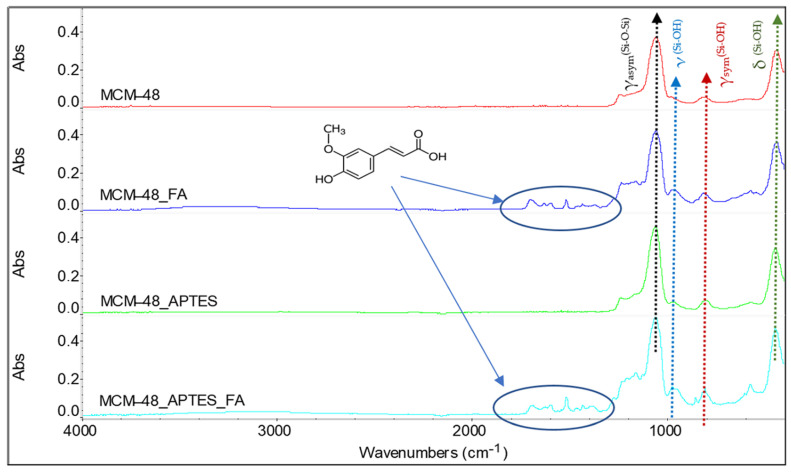
FTIR spectra and band assignments for the samples MCM-48, MCM-48_FA, MCM-48_APTES and MCM-48_APTES_FA.

**Figure 6 pharmaceutics-15-00660-f006:**
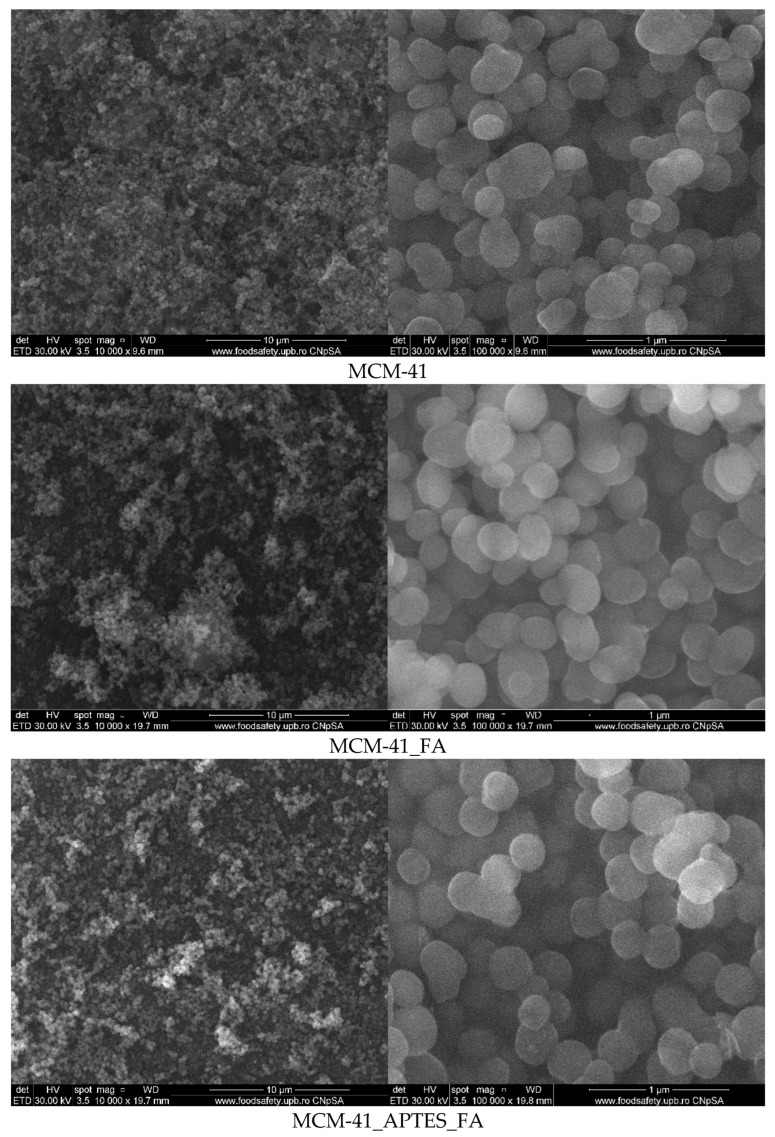
SEM micrographs for mesoporous materials based on MCM-41 support.

**Figure 7 pharmaceutics-15-00660-f007:**
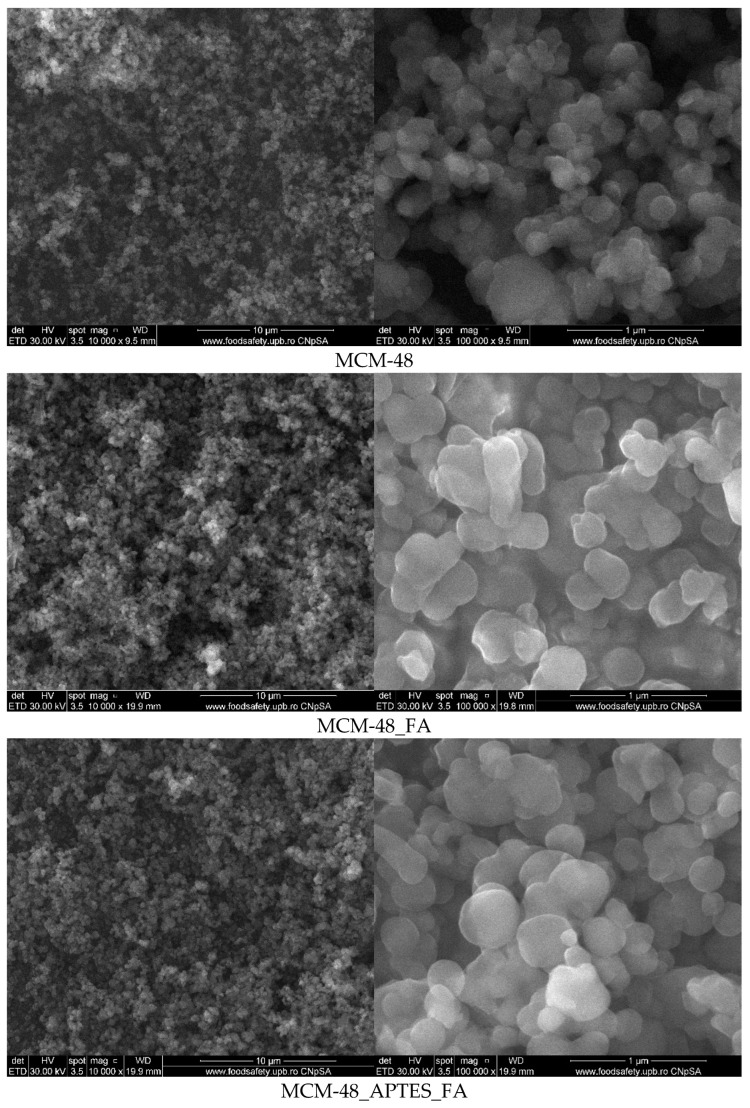
SEM micrographs for mesoporous materials based on MCM-48 support.

**Figure 8 pharmaceutics-15-00660-f008:**
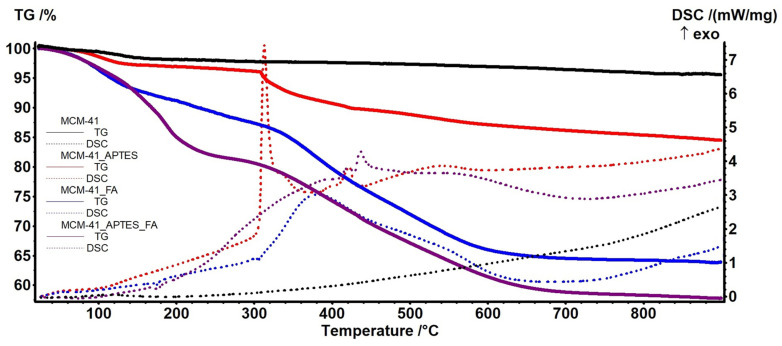
The TG (solid)—DSC (dotted) curves for the MCM-41 (black); MCM-41_APTES (red); MCM-41_FA (blue) and MCM-41_APTES_FA (purple).

**Figure 9 pharmaceutics-15-00660-f009:**
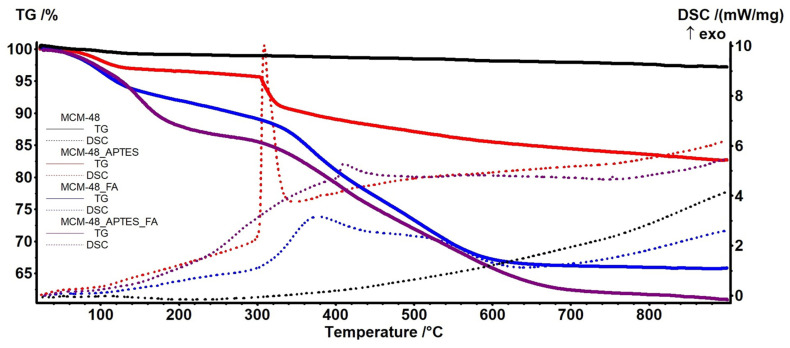
The TG (solid)—DSC (dotted) curves for the MCM-48 (black); MCM-48_APTES (red); MCM-48_FA (blue) and MCM-48_APTES_FA (purple).

**Figure 10 pharmaceutics-15-00660-f010:**
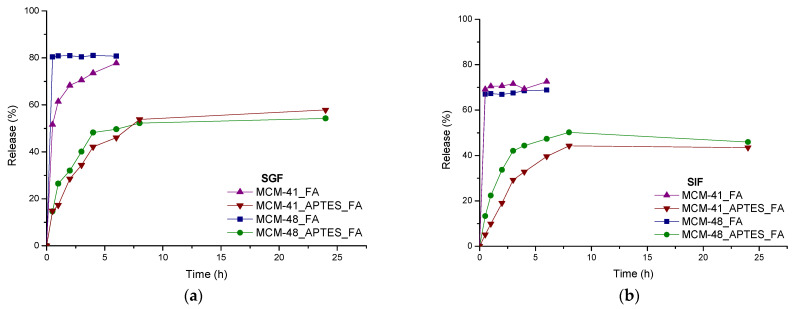
Release profiles for FA from the MCM-41_FA; MCM-41_APTES_FA; MCM-48_FA; MCM-48_APTES_FA samples in SGF (**a**) and SIF (**b**).

**Table 1 pharmaceutics-15-00660-t001:** Types of mesoporous material loaded with trans-ferulic acid (FA).

Sample Code	Materials Type
MCM-41	MCM-41
MCM-41_APTES	MCM-41: (3-aminopropyl)triethoxysilane
MCM-41_FA	MCM-41: trans-ferulic acid
MCM-41_APTES_FA	MCM-41: (3-aminopropyl)triethoxysilane: trans-ferulic acid
MCM-48	MCM-48
MCM-48_APTES	MCM-48: (3-aminopropyl)triethoxysilane
MCM-48_FA	MCM-48: trans-ferulic acid
MCM-48_APTES_FA	MCM-48: (3-aminopropyl)triethoxysilane: trans-ferulic acid

**Table 2 pharmaceutics-15-00660-t002:** BET characteristics of mesoporous materials.

Type of Material	BET Surface Area m^2^/g	Langmuir Surface Area m^2^/g	Volume of Pores cm^3^/g
MCM-41	1365	2.294	0.783
MCM-41_APTES	1014	2.251	0.5706
MCM-41_APTES_FA	301.3	2.564	0.1931
MCM-48	1582	2.383	0.9423
MCM-48_APTES	1555	1.897	0.7371
MCM-48_APTES_FA	504.6	2.218	0.2798

**Table 3 pharmaceutics-15-00660-t003:** Surface density of adsorbed H2O molecules and -OH moieties.

Sample	1st Mass Loss (%)	2nd Mass Loss (%)	n_H2O_ (mmol/g)	n_OH_ (mmol/g)	N_H2O_ (Groups/nm^2^)	N_OH_ (Groups/nm^2^)
MCM-41	1.83	2.61	1.02	2.90	0.45	1.28
MCM-48	0.94	1.97	0.52	2.19	0.20	0.83

**Table 4 pharmaceutics-15-00660-t004:** Most important data from thermogravimetric analysis.

Sample	Mass Loss RT-305 °C (%)	Mass Loss 305–700 °C (%)	Residual Mass (%)	Estimated FA Load (%)
MCM-41_APTES	3.99	9.94	84.49	-
MCM-41_FA	12.82	22.70	63.86	33.15
MCM-41_APTES_FA	19.55	21.65	57.74	31.66
MCM-48_APTES	4.47	11.19	82.65	-
MCM-48_FA	11.08	22.77	65.81	32.26
MCM-48_APTES_FA	14.63	22.98	60.90	26.32

**Table 5 pharmaceutics-15-00660-t005:** Determined values for minimum inhibitory concentration (MIC).

Strains	MIC (mg/mL)								
	MCM-41	MCM-41_ APTES	MCM-41_ FA	MCM-41_ APTES_FA	MCM-48	MCM-48_ APTES	MCM-48_ FA	MCM-48_ APTES_FA	C Ferulic Acid
*S. aureus* ATCC 25923	0.1	0.001	1	0.001	0.01	0.01	10	0.01	10
*E. coli* ATCC 25922	0.1	0.1	1	0.1	0.1	0.1	0.01	0.1	1
*P. aeruginosa* ATCC 27853	0.01	0.01	1	0.01	0.01	0.01	0.1	1	1
*C. albicans* ATCC 10231	1	0.01	1	0.001	0.01	0.1	0.1	1	100

**Table 6 pharmaceutics-15-00660-t006:** Determination of minimum adhesion inhibition concentration (MAIC).

Strains	MAIC (mg/mL)								
	MCM-41	MCM-41_ APTES	MCM-41_ FA	MCM-41_ APTES_FA	MCM-48	MCM-48_ APTES	MCM-48_FA	MCM-48_ APTES_FA	C Ferulic Acid
*S. aureus* ATCC 25923	0.1	0.001	0.1	0.001	0.01	0.01	1	0.01	1
*E. coli* ATCC 25922	0.1	0.1	1	0.1	0.1	0.1	0.01	0.1	0.1
*P. aeruginosa* ATCC 27853	0.01	0.01	1	0.01	0.01	0.01	0.1	1	1
*C. albicans* ATCC 10231	0.1	0.001	1	0.001	0.01	0.01	0.01	1	100

## Data Availability

Not applicable.

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
