# Peer review of "Increasing Bioavailability of Trans-Ferulic Acid by Encapsulation in Functionalized Mesoporous Silica"

_pharmaceutics, 2023, doi:10.3390/pharmaceutics15020660_

Round 1

Reviewer 1 Report

The paper presents interesting results on the fabrication and characterization of functionalized mesoporous silica for the controlled release of trans-ferulic acid. The paper can be accepted for publication after major revision.  The following issues should be clarified:

More comprehensive characteristics of mesoporous MCM-41 (hexagonal) and MCM-48 (cubic arrangement of the pores) should be added. It is hard to understand the differences between these two for non-specialists in this narrow area. 

The work does not specify whether a monolayer of APTES-modified mesoporous silica or whether multilayer structures were formed.

It is not clear from the text why, after the FA loading, the BET surface area drastically decreases. Appropriate discussion should be added.

Finally, I suggest citing papers where similar advanced drug delivery systems were described:

https://doi.org/10.3390/polym14194245

https://doi.org/10.3390/pharmaceutics14050909

Author Response

Reviewer 1

The paper presents interesting results on the fabrication and characterization of functionalized mesoporous silica for the controlled release of trans-ferulic acid. The paper can be accepted for publication after major revision.  The following issues should be clarified:

Answer:

We are grateful to the esteem reviewer for the appreciation words. Following the helpful advices received we further improved the manuscript and we hope that the esteem reviewer will find it suitable for publishing.

Point 1:

More comprehensive characteristics of mesoporous MCM-41 (hexagonal) and MCM-48 (cubic arrangement of the pores) should be added. It is hard to understand the differences between these two for non-specialists in this narrow area.

Response 1:

We are deeply grateful to the esteem reviewer for pointing out this weakness. We have added the missing information:

“The major difference between the two materials is related to the arrangements of the pores. If MCM-41 has the pores arranges uniaxial, the MCM-48 has the pores arranged in a three-dimensional fashion and thus, the release of the active components will be unidirectional for MCM-41 or in all directions for MCM-48”

Point 2:

The work does not specify whether a monolayer of APTES-modified mesoporous silica or whether multilayer structures were formed.

Response 2:

The method used allows formation of monolayer APTES-mesoporous silica. The small decrease of the BET surface also indicates a monolayer formation.

Point 3:

It is not clear from the text why, after the FA loading, the BET surface area drastically decreases. Appropriate discussion should be added.

Response 3:

We are grateful to the esteem reviewer for pointing out this weakness. We have enriched the section 3.2 “Specific surface area - Brunauer-Emmet-Teller adsorption isotherms” and we have discussed comparatively with other reported literature results.

“The overall surface area of these mesoporous materials has two components, the external surface area of the spherical particles, with a small contribution – less than 1 m2/g, and the internal surface area of the cylindrical pores, which assure the very high surface area. Because during the loading process, the FA will enter inside the pores of the material, filling them, the BET determined specific surface area will drastically decrease, roughly by a factor of 3. As a consequence of the FA loading, the pore size decrease, and some of them even disappear when the loading degree is high, therefore, the overall specific surface area de-creases considerable.”

Point 4:

Finally, I suggest citing papers where similar advanced drug delivery systems were described:

https://doi.org/10.3390/polym14194245

https://doi.org/10.3390/pharmaceutics14050909

Response 4:

We thank esteem reviewer for indicating this valuable studies. We have added comparison with them in relevant places

Reviewer 2 Report

This manuscript reported the fabrication and characterizations of two types of mesoporous silica loaded with trans-ferulic acid, the release behaviors and anti-bacterial activities of the two drug-loaded systems were investigated as well. Authors presented the preparation and physicochemical characterizations in detail of the two functionalized mesoporous silica materials, however, there are major concerns raised with the results of drug release properties and the activity of antibacterial.

(1)   In Figure 10, it was shown that the adsorbed FA was released almost completely within the first 5 h, which of course due to the weak binding with the mesoporous silica. However, it should be noticed that there were only about 50% of the drug released within the first 5 h when the compound was covalently bound to the mesoporous silica, and then, no drug was released after the first 5 h, therefore, the results could not support the conclusion of controlled release in long term. The question is where the rest amount of the drug? When and would the rest drug be released?

(2)   The encapsulation efficiency should be determined, which was missed in this study.

(3)   The novelty or specific features of this study requires clear demonstration, in the current manuscript, it is hard to tell.

Author Response

Reviewer 2

This manuscript reported the fabrication and characterizations of two types of mesoporous silica loaded with trans-ferulic acid, the release behaviors and anti-bacterial activities of the two drug-loaded systems were investigated as well. Authors presented the preparation and physicochemical characterizations in detail of the two functionalized mesoporous silica materials, however, there are major concerns raised with the results of drug release properties and the activity of antibacterial.

Response

Following the helpful advices received, we further improved the manuscript and we hope that the esteem reviewer will find it suitable for publishing.

Point 1:

In Figure 10, it was shown that the adsorbed FA was released almost completely within the first 5 h, which of course due to the weak binding with the mesoporous silica. However, it should be noticed that there were only about 50% of the drug released within the first 5 h when the compound was covalently bound to the mesoporous silica, and then, no drug was released after the first 5 h, therefore, the results could not support the conclusion of controlled release in long term. The question is where the rest amount of the drug? When and would the rest drug be released?

Response 1:

We are grateful to the esteem reviewer for pointing out this weakness. We have added the following explanation:

“Considering the gastrointestinal transit time, it can see that within the first 2-5 h most of the FA will be released in the stomach follower by 2-6 h of release is small intestines and 10-60 h in the large bowel so, most probably the overall release will be more than 40-60%. Moreover, these systems can be embedded in mucoadhesive polymers to improve the residence time and recovery %. In addition, comparing with antimicrobial drugs, such as antibiotics, the negative environmental impact is low as these systems don’t lead to antimicrobial resistance.”

Point 2:

The encapsulation efficiency should be determined, which was missed in this study.

Response 2:

The encapsulation efficiency is practically 100% as we used vacuum assisted loading method in which all the FA from acetone solution will be found in the final dry product.

“The as obtained functionalized mesoporous materials were loaded with trans-ferulic acid using the vacuum assisted loading technology and found that the loading mainly occurs within the pores”

Point 3:

The novelty or specific features of this study requires clear demonstration, in the current manuscript, it is hard to tell.

Response 3:

We are thankful for the time and effort spent to indicate the areas that needed to be improved. We have enriched the article with more data and comparison with literature in order to demonstrate the novelty of this research.

As a side note we have searched the Clarivate database (with EndNote) with “ferulic acid” and “silica” as keywords for title and the returned hits consists of 5 articles, none of them about APTES functionalized MCM-41 or MCM-48, and of course none with vacuum assisted loading method. Therefore, we can state that this is the first comparative study on MCM-41/48 and APTES functionalized MCM-41/48 loaded with FA.

Round 2

Reviewer 1 Report

The authors have answered all my comments and the paper can be accepted in its present form.

Author Response

We are thankful to the esteem reviewer for the time and effort spent to indicate the areas that needed to be improved.

Reviewer 2 Report

Authors have made responses to most of the concerns. However, as for the response that "Considering the gastrointestinal transit time, it can see that within the first 2-5 h most of the FA will be released in the stomach follower by 2-6 h of release is small intestines and 10-60 h in the large bowel so, most probably the overall release will be more than 40-60%. Moreover, these systems can be embedded in mucoadhesive polymers to improve the residence time and recovery %. In addition, comparing with antimicrobial drugs, such as antibiotics, the negative environmental impact is low as these systems don’t lead to antimicrobial resistance.”, there are still several points required further explanation:

(1) Are the explanation for the release behaviors derived from experimental data, or authors assumption, or from the cited reference 63? It can not see the release behaviors or biodistributions in the different tissues from the experimental data. Please clarify in the text.

(2) What did the "mucoadhesive polymers" mean? Are there any relavant referecnes to support? please explain.

(3) "follower" should be "following"?

Author Response

Authors have made responses to most of the concerns. However, as for the response that "Considering the gastrointestinal transit time, it can see that within the first 2-5 h most of the FA will be released in the stomach follower by 2-6 h of release is small intestines and 10-60 h in the large bowel so, most probably the overall release will be more than 40-60%. Moreover, these systems can be embedded in mucoadhesive polymers to improve the residence time and recovery %. In addition, comparing with antimicrobial drugs, such as antibiotics, the negative environmental impact is low as these systems don’t lead to antimicrobial resistance.”, there are still several points required further explanation:

Response

Following the helpful advices received, we further improved the manuscript and we hope that the esteem reviewer will find it suitable for publishing.

Point 1:

(1) Are the explanation for the release behaviors derived from experimental data, or authors assumption, or from the cited reference 63? It can not see the release behaviors or biodistributions in the different tissues from the experimental data. Please clarify in the text.

Response 1:

We are grateful to the esteem reviewer for pointing out this issue and giving us the opportunity to further clarify the statement. We have modified the paragraph as follows:

“As indicated by the experimental data presented in [63] the normal gastrointestinal transit time can be divided in three intervals: first 2-5 h are spent in the stomach, followed by 2-6 h of release in small intestine and 10-60 h in the large bowel. By coupling these times with the data obtained from release profiles in SGF and SIF, we can state that between 20-40% of FA will be released in the stomach (depending on actual transit time), with additional 20-40% of loaded FA being release in the small intestine. Additional FA quantities can be released in large bowel, function of the actual transit time.”

Point 2:

What did the "mucoadhesive polymers" mean? Are there any relavant referecnes to support? please explain.

Response 2:

The mucoadhesive polymers, as their name suggest, can adhere to the mucus or to the cell membrane due to their multiple hydrophilic moieties like hydroxyl, carboxyl, amide and sulfate. It can protect the loaded drug function of pH. Once it adheres to the mucus, the active substance can be released longer time that the simple intestinal transit allows.

We have modified the paragraph and added relevant references:

In addition, after FA loading, the APTES functionalized MCM-41 or MCM-48 particles can be embedded in a mucoadhesive system [64,65], that will allow modulation of residence time and FA recovery [66].

Point 3:

"follower" should be "following"

Response 3:

We are thankful for the time and effort spent to indicate the areas that needed to be improved. We have made the necessary correction.